# Oxidoreductase gene *fabG* contributes to fungal development, cell wall integrity, and virulence in *Aspergillus fumigatus*

Heng Zhang,[1] Hua Ni,[2] Xinyi Tao,[3] Tian Chen,[3] Yi Zhang,[4] Xiaolei Zhu,[1] Yinping Chen,[1] Mengqi Peng,[1] Yi Sun[1]

**ABSTRACT** *Aspergillus fumigatus*, a major cause of invasive aspergillosis, relies on oxidoreductases for stress adaptation. The role of the oxidoreductase gene *fabG* in fungal physiology and virulence remains unclear. This study aims to investigate the function of *fabG* in regulating *A. fumigatus* growth, virulence, redox homeostasis, and cell wall integrity (CWI). *fabG* knockout (Δ*fabG*) and complemented strains were constructed via homologous recombination. Phenotypic assays evaluated hyphal growth, virulence in *Galleria mellonella*, and antifungal susceptibility. Transcriptomic profiling, cell wall composition analysis (chitin/β-glucan), biochemical assays (superoxide dismutase [SOD] activity, reactive oxygen species [ROS] levels, and stress tolerance), and macrophage co-culture experiments (phagocytosis/killing) were performed. CWI-related gene expression was assessed under caspofungin/rapamycin treatment. Δ*fabG* showed inhibited hyphal growth, reduced virulence, and decreased caspofungin sensitivity. Transcriptomics revealed altered oxidoreductase and mitogen-activated protein kinase signaling genes. Cell wall thickening (increased chitin) and enhanced tolerance to cell wall stressors indicated CWI activation. Δ*fabG* exhibited decreased SOD activity, elevated ROS, reduced stress tolerance, and increased susceptibility to macrophage killing. CWI-related gene expression was unchanged under caspofungin/rapamycin. *fabG* critically regulates *A. fumigatus* growth, environmental adaptation, and virulence by modulating CWI and redox homeostasis, highlighting its potential as an antifungal target.

**IMPORTANCE** *Aspergillus fumigatus* is a harmful fungus that can cause severe and often deadly infections in people with weakened immune systems. Understanding how this fungus adapts to stress and causes disease is crucial for developing new treatments. Our study reveals that the gene *fabG* plays a vital role in regulating the growth, stress response, and virulence of *A. fumigatus*. When FabG is missing, the fungus becomes less able to grow, resists environmental stresses poorly, and is less harmful to infected hosts. This gene controls the fungus's ability to maintain internal balance during oxidative stress and strengthens its cell wall, a protective structure. Since FabG is critical for the fungus's survival and virulence, it could be a promising target for new antifungal drugs. By blocking *fabG*, we might be able to develop more effective treatments against this dangerous pathogen.

**KEYWORDS** *fabG*, *Aspergillus fumigatus*, oxidoreductase, cell wall integrity, virulence

The genus *Aspergillus*, comprising filamentous fungi ubiquitously distributed in soil, decaying organic matter, and air, includes *Aspergillus fumigatus* as one of the most virulent human pathogens (1, 2). This species thrives due to its unique biological traits, such as thermotolerance, robust oxidative stress adaptation, and efficient spore dispersal (3–8).

**Peer Reviewer** Hee-Soo Park, Kyungpook National University, Daegu, Republic of Korea

Address correspondence to Yi Sun, jzzxyysy@163.com.

Heng Zhang and Hua Ni contributed equally to this article. The author order was determined based on their contribution to the article.

The authors declare no conflict of interest.

See the funding table on p. 19.

The rising prevalence of immunosuppressive therapies and organ transplantation has dramatically expanded the population at risk for invasive aspergillosis (IA). Epidemiological data indicate that approximately 30 million individuals worldwide are threatened by IA, with over 1 million new cases annually and mortality rates reaching 30%–90% (9, 10). Chronic pulmonary aspergillosis further underscores its clinical burden, exhibiting a 5-year survival rate of only 62% (11, 12). With a growing at-risk population, these infections are of critical concern, resulting in *A. fumigatus* being listed by the WHO as a critical priority fungal pathogen.

The pathogenicity of *A. fumigatus* hinges on its ability to counteract host microenvironmental stresses. During host-pathogen interactions, the fungus must neutralize reactive oxygen species (ROS) generated by immune cells (e.g., neutrophils and macrophages) while maintaining intracellular redox homeostasis (13, 14). Oxidoreductases play pivotal roles in this process: they scavenge ROS (e.g., superoxide dismutase [SOD] and catalase [CAT]) to mitigate oxidative damage and regulate critical metabolic pathways (e.g., ergosterol biosynthesis) and virulence factor expression (15). Concurrently, the cell wall integrity (CWI) pathway dynamically modulates the synthesis and repair of cell wall components (e.g., β-1,3-glucan and chitin). For instance, CYP51 (14-α sterol demethylase), the target of azole antifungals, is involved in membrane integrity by catalyzing sterol precursor demethylation; its inhibition triggers ROS accumulation and fungal death (16). However, the molecular interplay between redox equilibrium and antifungal efficacy remains poorly understood, and evolving drug resistance further complicates clinical management.

Notably, *A. fumigatus* survival and virulence require not only redox homeostasis but also coordinated activation of the CWI pathway to withstand external pressures. The CWI pathway senses cell wall damage and activates downstream signaling cascades (e.g., mitogen-activated protein kinase [MAPK] such as Mkk2), thereby regulating β-1,3-glucan synthesis and repair (17). Echinocandins (e.g., caspofungin [CAS]) disrupt fungal cell wall structure by inhibiting β-1,3-glucan synthase. Although their efficacy may be partially limited by compensatory activation of the CWI pathway, they are still commonly used as part of combination therapy or as a second-line option in the treatment of IA (18, 19). Emerging studies suggest that in various species, there is crosstalk between the redox system and the CWI signaling pathway: elevated ROS levels can activate the CWI pathway, initiating the repair of cell wall defects caused by oxidative damage (20–24). Therefore, it is reasonable to speculate that aberrant activation of the CWI pathway may, in turn, regulate the expression of antioxidant enzymes (such as SOD), forming a complex adaptive regulatory network. Nevertheless, how oxidoreductases (e.g., the Oxidoreductase family) collaborate with the CWI pathway to regulate pathogenicity and drug responses in *A. fumigatus* remains unresolved. Although knowledge of genes involved in cell wall (CW) biosynthesis is expanding, over 30% of the *A. fumigatus* genome still encodes genes with uncharacterized functions (25).

In this preliminary study, we employed homologous recombination to replace the *fabG* gene with a *pyrG* marker, generating a *fabG* knockout strain (Δ*fabG*) (26). This model was used to investigate the impact of *fabG* deletion on fungal growth, drug susceptibility, and virulence, as well as its role in oxidative stress and CWI signaling. Our findings unveil novel functions of *fabG* in *A. fumigatus*, providing critical insights into its adaptive mechanisms and therapeutic potential.

## MATERIALS AND METHODS

### *A. fumigatus* strains, media, and growth conditions

In this study, the *A. fumigatus* KU80 strain (a uracil auxotroph strain lacking *pyrG* and *akuB* genes, Fungal Genetics Stock Center) was used as the wild-type genetic background to construct the *fabG* gene deletion strain (Δ*fabG*) and its complemented strain (Δ*fabG::fabG*⁺) (27, 28).

The media used in this study included SAB, CZA, LB, and RPMI 1640. All liquid media were prepared from the corresponding solid media without agar. Fresh *A. fumigatus* conidia were harvested from solid SAB plates with sterile water and adjusted to the required concentration for each experiment using a hemocytometer. Unless otherwise specified, strains were incubated at 37°C.

## Phylogenetic tree construction

The sequences of *fabG* orthologs were downloaded from the NCBI (https://www.ncbi.nlm.nih.gov) and FungiDB (http://fungidb.org/fungidb) websites. A phylogenetic tree was constructed using the neighbor-joining method with MEGA 10 software, and sequence alignment was conducted using Cluster W. Conserved domain analysis of proteins was performed using the Conserved Domain Database website on the NCBI. ESPript (https://espript.ibcp.fr/ESPript/ESPript) was used for multiple sequence alignment.

## Generation of mutant strains

*A. fumigatus* KU80 was used as the parental isolate to generate knockout mutants. Knockout cassettes were generated using the protocol from Zhao et al. (26) (Fig. S1). Briefly, 1.2-kb flanking regions of the gene of interest were PCR amplified and fused to a *pyrG* cassette via additional fusion PCR. The *A. fumigatus* KU80 strain was cultured overnight at 37°C and 120 rpm in CZA liquid medium supplemented with uracil, followed by a 5-hour protoplasting treatment in Sabouraud agar + protoplasting solution (pectinase + freshly filtered 0.6 M KCl, citric acid). Protoplasts were filtered through Miracloth (Sigma, USA), washed twice in 0.6 M KCl, and resuspended in 0.6 M KCl + 200 mM $CaCl_2$. Fusion PCR product was added to $1 \times 10^5$ protoplasts, followed by addition of PEG. This was incubated on ice for 30 minutes. In total, 200 µL of PEG was added, and the mixture was then incubated at room temperature for 10 minutes. The transformation mixture was plated on CZA agar. Transformants were purified twice on CZA agar and PCR validated (Fig. S2). All primers used in this study are listed in Table S1.

## Construction of the *fabG* complemented strain

The *fabG* gene was amplified from the genomic DNA of *A. fumigatus* using primers Aim-Re-F and Aim-Re-R in a PCR experiment. The amplified *fabG* gene was subsequently subcloned into the NaeI and KpnI sites of the plasmid PCT74, resulting in the recombinant plasmid p-hph-fabG. The constructed plasmid was transformed into the Δ*fabG* deletion strain to generate the Δ*fabG* complemented strain (Δ*fabG::fabG*+) (Fig. S1). All primers used in this experiment are listed in Table S1.

## E-TEST

*A. fumigatus* conidia suspension ($2 \times 10^5$ cfu/mL) was evenly spread on the surface of RPMI 1640 agar plates (containing 2% agar, pH 7.0). The E-TEST strips were then gently placed on the agar surface, ensuring full contact with the medium. Plates were incubated at 37°C for 48 hours. After incubation, the lowest drug concentration at the intersection of the inhibition zone (elliptical area) with the E-TEST strip was recorded as the minimum effective concentration (MEC).

## M38-A3 broth microdilution method

The broth microdilution method was performed according to the M38-A3 standard established by the Clinical Laboratory Standard Institute, using RPMI 1640 liquid medium (29). Antifungal drugs (voriconazole [VOR], itraconazole [ITR], posaconazole [POS], and CAS) were prepared in eight twofold serial dilutions, with the highest concentration of 8 µg/mL and a working concentration range of 0.25–8 µg/mL. A 96-well microplate was used, with 100 µL of each drug dilution added to each well, followed by 100 µL

of fungal suspension to achieve a final fungal concentration of $2 \times 10^4$ cfu/mL. Wells without drugs were included as growth control. The plates were incubated at 35°C for 48 hours, and the results were recorded after incubation. To ensure accuracy, the experiment was performed in triplicate. *Candida parapsilosis* ATCC22019 and *Aspergillus flavus* ATCC204304 were used as the quality control strain. In addition, the same method was used to test cell wall stressors (calcofluor white [CFW] and Congo red [CR]) and oxidative stress-inducing agents (menadione and $H_2O_2$). These agents were prepared in 12 twofold serial dilutions. The maximum concentration of hydrogen peroxide was 10%, with a working concentration range of 0.002%–10%, while the maximum concentrations of the other agents were 800 µg/mL, with a working concentration range of 0.25–800 µg/mL.

## Radial growth germination rate analysis

Referring to Michael J. Bromley's method (28), 500 µL of $5 \times 10^5$ cfu/mL suspension was inoculated into RPMI-1640 medium with 2.0% glucose and 165 mM MOPS buffer (pH 7.0) in a 24-well glass-bottom plate. The culture was incubated at 37°C, and the optical density at 600 nm was measured using a multifunctional microplate reader (Allsheng, China).

## Spore quantification assay

A layer of cellophane was placed on solid SAB medium, and 1 µL of spore suspension ($5 \times 10^5$ cfu/mL) was inoculated in the center. After drying, it was incubated at 37°C for 3 days, with spore production measured daily. Following incubation, the cellophane was carefully removed, and sporangiophores were scraped off completely with a swab. Conidia were observed under a bright field microscope and counted with a hemocytometer.

## Sensitivity assays for oxidative stress

Congo red (200 µg/mL) and calcofluor white (100 µg/mL) were used to assess CWI; mannitol (2 M), sorbitol (1 M), and NaCl (1 M) were used for osmotic stress sensitivity testing; hydrogen peroxide $H_2O_2$ (5 M) and menadione (200 µM) were used to evaluate oxidative stress responses. All plates were prepared using SAB medium as the base. A 2 µL conidial suspension ($1 \times 10^6$ cfu/mL) was spotted onto the center of each plate. Plates were incubated at 37°C, and colony growth was photographed after 48 hours. Each experiment was performed in triplicate with biological replicates. The inhibition rate was calculated as follows: inhibition rate = (colony diameter without stressors − colony diameter with stressors)/colony diameter without stressors × 100%. "Colony diameter without stressors" specifically refers to the colony diameter of strains cultured on solid SAB medium at 37°C for 48 hours, serving as a unified baseline for inhibition rate calculation across all stress-related assays.

## Cell wall thickness analysis

One milliliter of conidia suspension of fresh strains at a concentration of $10^8$ cfu/mL was inoculated into 100 mL of liquid SAB medium and incubated at 37°C and 220 rpm for 24 hours. A portion of the mycelium was taken and soaked in glutaraldehyde and sent to the Wuhan Service Biotechnology, China, for scanning transmission electron microscopy. Cell wall thickness was determined using ImageJ 1.54 software, with measurements taken at three random locations on the cell wall and averaged to obtain the final thickness value (30).

## β-Glucan assay

The β-glucan assay was performed following methods previously described (31, 32). Briefly, $1 \times 10^7$ conidia of each strain were grown overnight in 25 mL of SAB broth at 37°C.

After 16 hours of growing, hyphae were collected by filtration through Miracloth (Sigma, USA) and washed using 0.1 M NaOH solution. Washed fungal hyphae were lyophilized for 24 hours. Five milligrams of dry hyphae were disrupted in a bead-beater three times (1 minute each) with 1 minute of ice incubation between each. Hyphal powder was resuspended to a final concentration of 20 mg/mL in 1 M NaOH, and the solution was incubated at 52°C for 30 minutes. Fifty microliters of each sample was mixed with 185 µL of aniline blue staining solution (183 mM glycine, 229 mM NaOH, 130 mM HCl, and 618 mg/L aniline blue, pH 9.9) into a 96-well masked fluorescence plate (Shenggong, China). The sample-containing plates were incubated at 52°C for 30 minutes, followed by a cool-down period of 30 minutes at room temperature. Fluorescence readings were performed using an excitation/emission wavelength of 405/460 nm, respectively (Allsheng, China). All the experiments were performed in triplicate using three independent *A. fumigatus* cultures, and the results were represented as relative quantification versus the WT strain.

## RNA-seq analysis and real-time quantitative PCR

Fresh *A. fumigatus* conidia were grown in liquid SAB in a rotary shaker at 220 rpm at 37°C for 48 hours. For RNA sequencing (RNA-seq) analysis, mycelial pellets were collected and quickly frozen in liquid nitrogen. After mRNA extraction, purification, and library construction, sequencing was performed by next-generation sequencing based on the Illumina sequencing platform. Genes were considered differentially expressed if they met both criteria: false discovery rate (FDR) < 0.01 and absolute $\log_2$ fold change ≥ 1.0. The detailed procedures were performed by Beijing Biomaker Biotechnology Co., Ltd. (China). Each sample was analyzed using three biological repetitions.

For RT-qPCR analysis, total RNA was extracted with the TRIeasy Total RNA Extraction Reagent (Yeasen, China) according to the manufacturer's directions. The Hifair III 1st Strand cDNA Synthesis SuperMix for qPCR (Yeasen, China) was used to synthesize cDNA. Independent assays were performed with three replicates, and transcript levels were calculated by the comparative threshold cycle (ΔCT) and normalized against the mRNA expression of *tubA* in *A. fumigatus*. The $2^{-\Delta\Delta CT}$ method was used to determine the changes in mRNA expression (33). All the RT-qPCR primers are given in Table S2.

## SOD and CAT activity measurement

To measure total SOD and CAT activity, $2 \times 10^7$ spores were inoculated into 100 mL of liquid SAB medium and shaken at 220 rpm and 37°C for 16 hours. After centrifugation at 12,000 *g* for 3 minutes, the pellet was washed twice with PBS. Small glass beads were added, and the mixture was homogenized in a high-speed oscillator (Benchmark, USA). The supernatant, collected after centrifugation at 8,000 *g* for 3 minutes, was used as the sample. Protein concentration was determined with a BCA Protein Quantification Kit (Yeasen, China). SOD activity was measured using a Total Superoxide Dismutase Activity Colorimetric Assay Kit II (Yeasen, China), and CAT activity was assessed with a Catalase Assay Kit (Beyotime, China) (15, 34).

## Morphological examination

Following the previously described method (35, 36), slide cultures were prepared and incubated at 37°C for 24 hours. Slide cultures were prepared and incubated at 37°C for 24 hours. Samples were then stained with CFW (Sigma, USA) and observed under a Leica DMiL LED fluorescence microscope (Leica, Germany).

## 2,7-dichlorofluorescin diacetate staining

Fresh conidia collected from SAB agar plates after 3 days of growth were transferred into SAB liquid medium and incubated at 37°C with 130 rpm shaking for 36 hours. 2,7-dichlorofluorescin diacetate (Yeasen, China; 10 µM) was then added to the sample at a volume of 0.1%. The mixture was incubated at 37°C for 30 minutes, followed by

centrifugation at 4,000 rpm for 20 minutes (37). Flow cytometry data were generated using a Beckman Cytomics FC 500 BD FACSCanto II and analyzed with FlowJo version 10 software. The excitation wavelength was 488 nm, and the emission wavelength was 525 nm.

### *Galleria mellonella* virulence assay

The *G. mellonella* virulence assay was performed using a method described previously (38). Fresh conidia of the corresponding strains were harvested and adjusted to $1 \times 10^8$ cfu/mL. Ten microliters of conidia suspension was then injected into the *G. mellonella* larvae through the left prolegs. The control group was injected with a sterile PBS solution. *G. mellonella* larvae were then incubated in darkness at 37°C for 7 days, and the number of larval deaths was recorded every 24 hours to calculate the survival rate. The log-rank (Mantel-Cox) test was used to compare survival curves. A *P* value of < 0.05 was considered statistically significant. Each experiment was replicated three times independently, and each replication contained 20 larvae per strain.

### Phagocytosis and fungal killing assay of *A. fumigatus* conidia by RAW 264.7 macrophage cells

RAW 264.7 macrophage cell line (Procell Life Science Co., Ltd., China) was used for the experiments. The cells were cultured in RPMI 1640 medium supplemented with 10% fetal bovine serum, 0.05 mM β-mercaptoethanol, and 1% penicillin (10,000 U/mL) and streptomycin (10,000 mg/mL) at 37°C in a humidified incubator with 5% $CO_2$. After adherence, RAW 264.7 cells were co-incubated with *A. fumigatus* conidia at 37°C and 5% $CO_2$ for 2 hours. Unphagocytosed conidia were gently washed away with PBS three times, and fresh medium was added. The cells were returned to the incubator and further incubated for 4 hours to assess fungal killing, or for 16 hours for morphological observation of the conidia. After 4 hours of incubation, the medium was removed, and the cells were collected and lysed by vigorous vortexing with small glass beads to release the internalized conidia. The resulting conidial suspension was serially diluted and immediately plated on SAB agar plates. After incubation under appropriate conditions, colony-forming units were counted, and the fungal killing rate by macrophages was calculated based on CFU data to evaluate the antifungal activity of RAW 264.7 cells. Meanwhile, total RNA was extracted from the cell suspension using Trizol reagent, and the RNA samples were stored at −80°C for further analysis. For microscopic observation of phagocytosed conidia, the PBS used for washing was supplemented with CFW (final concentration: 5 µg/mL), and the samples were incubated at 37°C for 10 minutes. The stained conidia were then observed under a fluorescence microscope using a hemocytometer.

### Statistical analysis

All the assays were done in triplicate on three independent days. All the statistical analyses were carried out using GraphPad Prism version 10.1.2 (GraphPad Software Inc., San Diego, CA, USA) for Windows. In each assay, at least three biological replicates were done to measure each parameter in each condition, avoiding biased results. Error bars included in all the graphs represent the standard deviation. ANOVA or *t*-test was used depending on whether we did multiple comparisons or compared punctual data, respectively, after ensuring that the data sets followed a normal distribution.

## RESULTS

### Identification of *fabG* in *A. fumigatus*

The FabG protein (corresponding to AFUA_8g01550) of *A. fumigatus* is conserved with respect to homologs in other species of the genus *Aspergillus* (Fig. 1A and C). The *fabG* gene is 808 bp in length and encodes a predicted protein of 250 amino acids. The

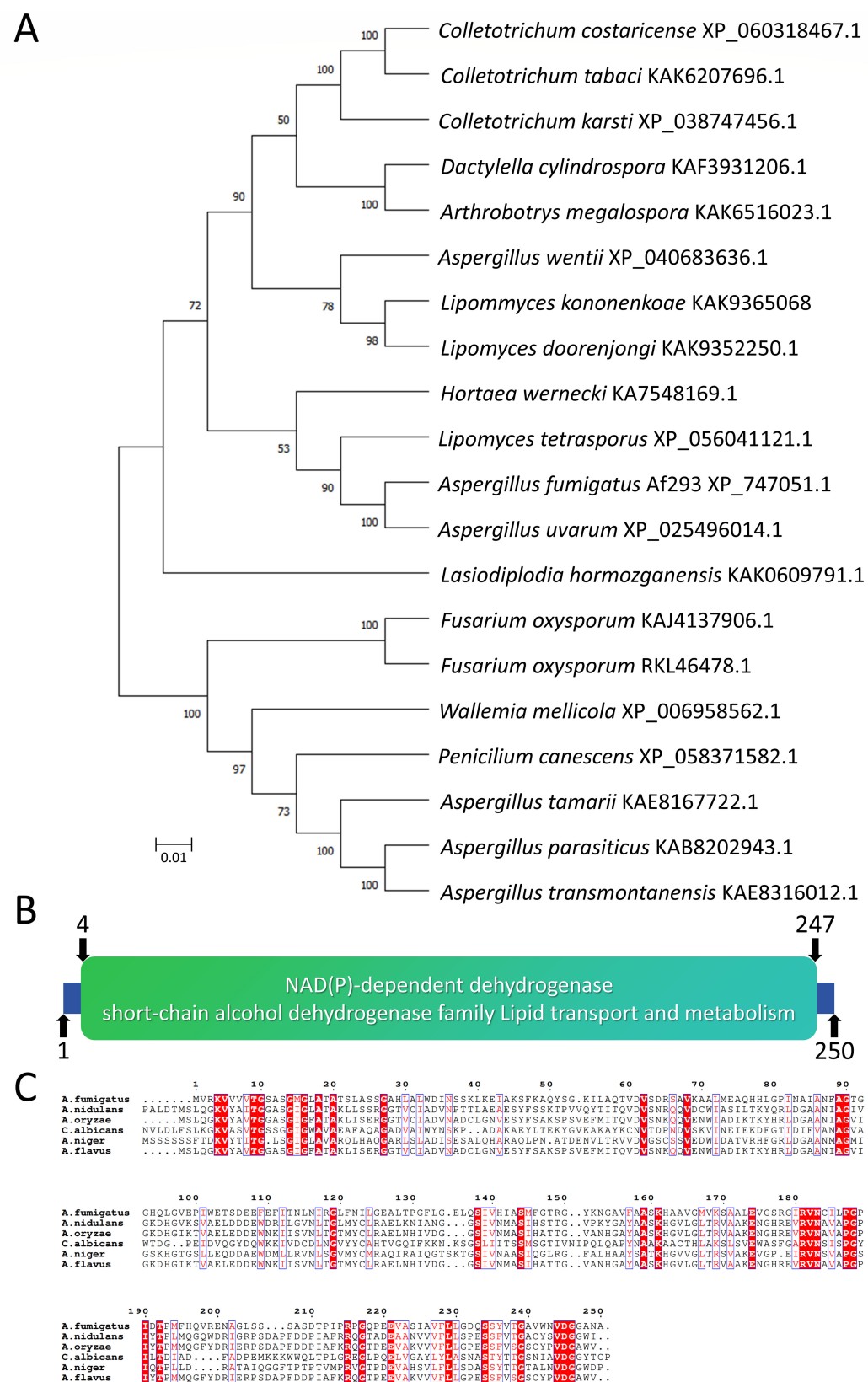

**FIG 1** *fabG* is conserved among common pathogenic filamentous fungi. (A) Phylogenetic tree of *fabG* homologs from different fungal species. Construction of a phylogenetic tree was carried out using MEGA version 11.0. The tree was generated with the maximum likelihood model with a bootstrap value of 1,000 (http://megasoftware.net). *Colletotrichum costaricense*, GenPept accession no. XP_060318467.1; *Colletotrichum tabaci*, GenPept accession no. KAK6207696.1;

Fig 1 (Continued)

*Colletotrichum karsti*, GenPept accession no. XP_038747456.1; *Dactylella cylindrospora*, GenPept accession no. KAF3931206.1; *Arthrobotrys megalospora*, GenPept accession no. KAK6516023.1; *Aspergillus wentii*, GenPept accession no. XP_040683636.1; *Lipomyces kononenkoae*, GenPept accession no. KAK9365068; *Lipomyces doorenjongi*, GenPept accession no. KAK9352250.1; *Hortaea wernecki*, GenPept accession no. KA7548169.1; *Lipomyces tetrasporus*, GenPept accession no. XP_056041121.1; *Aspergillus fumigatus* Af293, GenPept accession no. XP_747051.1; *Aspergillus uvarum*, GenPept accession no. XP_025496014.1; *Lasiodiplodia hormozganensis*, GenPept accession no. KAK0609791.1; *Fusarium oxysporum*, GenPept accession no. KAJ4137906.1; *Fusarium oxysporum*, GenPept accession no. RKL46478.1; *Wallemia mellicola*, GenPept accession no. XP_006958562.1; *Penicillium canescens*, GenPept accession no. XP_058371582.1; *Aspergillus tamarii*, GenPept accession no. KAE8167722.1; *Aspergillus parasiticus*, GenPept accession no. KAB8202943.1; *Aspergillus transmontanensis*, GenPept accession no. KAE8316012.1. (B) Display of the amino acid domains of FabG. (C) Sequence alignment of FabG using an online alignment tool (https://espript.ibcp.fr/ESPript/ESPript).

protein contains a domain from positions 4 to 247: NAD(P)-dependent dehydrogenase, short-chain alcohol, dehydrogenase family, region_name as *fabG* (Fig. 1B). It belongs to the short-chain dehydrogenases/reductases family, a diverse family of oxidoreductases with a single domain. These enzymes catalyze a wide range of activities, including the metabolism of steroids, cofactors, carbohydrates, lipids, aromatic compounds, and amino acids, and are involved in redox sensing. BLAST analysis showed that the closest homolog to *A. fumigatus* FabG was found in *Aspergillus uvarum* (Fig. 1A).

## Loss of *fabG* causes severe growth inhibition

In both solid media (CZA and SAB Agar) and liquid culture systems (RPMI-1640 and SAB broth), the Δ*fabG* mutant displayed drastically reduced conidiation and hyphal development compared to the WT and Δ*fabG::fabG*+ (Fig. 2A through D). Specifically, after cultivation in CZA for 24, 48, and 72 hours, the colony diameters of Δ*fabG* were reduced by 55.7%, 59.0%, and 52.9%, respectively, compared to WT (Fig. 2E). In SAB media, the reduction was 61.6%, 70.8%, and 73.4% (Fig. 2F). Similar results were observed in Δ*fabG::fabG*+ (Fig. 2F). In addition, Δ*fabG* produced significantly fewer spores than WT and Δ*fabG::fabG*+, with no conidia after 24 hours (Fig. 2G). The germination rate assay showed delayed spore germination in Δ*fabG* (Fig. 2H). Under optical microscopy, after 48 hours, WT and Δ*fabG::fabG*+ strains developed numerous conidiophores and began to produce conidia. However, after 72 hours, only hyphal tip swelling was observed in Δ*fabG*, and the conidiophores had not matured (Fig. 2I). This indicates that the deletion of *fabG* severely impairs hyphal growth and conidiation ability. Together, these results suggested that *fabG* is necessary for hyphal growth and conidiation in *A. fumigatus*.

## Antifungal drug sensitivity

Azoles and echinocandins are commonly used to treat IA, and the sensitivity of *A. fumigatus* to these antifungal agents is often a critical determinant of treatment efficacy. To further investigate the sensitivity of the Δ*fabG* mutant to these antifungal drugs, we evaluated its response to multiple antifungal agents. The results showed that, compared to the wild-type strain, Δ*fabG* exhibited reduced *s*ensitivity only to CAS, while no significant changes in sensitivity to azoles were observed. Since CAS targets fungal cell wall synthesis, and Δ*fabG* also exhibited higher resistance to cell wall stressors (CFW and CR), this suggests that FabG may be associated with genes involved in cell wall biosynthesis. Notably, Δ*fabG* also displayed reduced sensitivity to oxidative stress-inducing agents such as menadione and $H_2O_2$ (Table 1; Fig. 3).

## *fabG* downregulates genes related to oxidoreductases and the MAPK pathway

To explore the mechanism by which *fabG* mediates hyphal growth and redox balance, we conducted RNA-seq analysis of WT and Δ*fabG* strains grown in SAB medium. A total of 1,145 genes were significantly differentially expressed (log2 fold change ≥ 1.0 or ≤−1.0, FDR < 0.01) in the Δ*fabG* mutant compared to the WT, with 358 genes upregulated and 787 genes downregulated (Fig. 4A). Kyoto Encyclopedia of Genes and Genomes (KEGG)

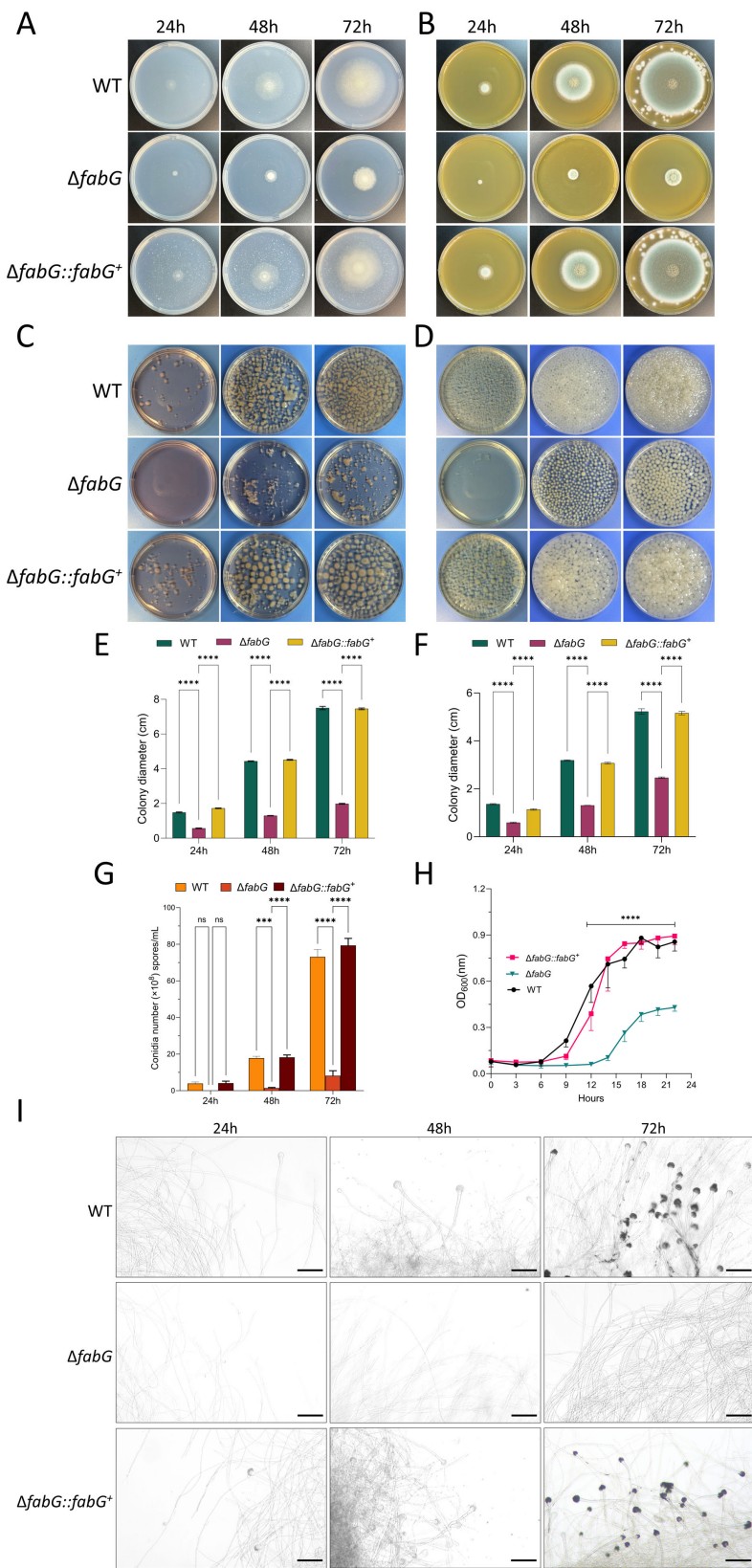

**FIG 2** *A. fumigatus fabG* is required for vegetative growth and conidiation. Representative growth phenotypes of the WT, Δ*fabG*, and Δ*fabG::fabG*⁺ on CZA (A and E), SAB (B and F) medium, RPMI-1640 (C), and SAB broth (D) at 37°C for 24, 48, and 72 hours. Diameters between strains were compared at (Continued on next page)

Fig 2 (Continued)

each time point using two-way ANOVA with Tukey's test for multiple comparisons. (G) Spore production numbers after growth on SAB medium for 1, 2, and 3 days. Statistical analyses were performed by two-way ANOVA with Tukey's test for multiple comparisons. (H) Radial growth germination rate analysis. *P*-value was calculated by repeated measures two-way ANOVA with Sidak's correction. (I) Mycelial morphology under optical microscopy, showing that Δ*fabG* lacks obvious conidiophores, particularly after 72 hours of cultivation. In contrast, the WT and Δ*fabG::fabG*⁺ strains exhibit prominent conidiophores and conidia production. Scale bar = 50 μm. ns, $P > 0.05$; ***$P < 0.001$; and ****$P < 0.0001$.

pathway enrichment analysis of all differentially expressed genes in the Δ*fabG* mutant revealed that the most significant pathways included Peroxisome, Mitogen-Activated Protein Kinase signaling pathway, and Biosynthesis of amino acids (Fig. 4B). Notably, within the altered Peroxisome-related genes, the expression of SOD-encoding gene (AFUA_5G09240) and Catalase-encoding gene (AFUA_2G18030) was significantly decreased. Additionally, the expression of the key regulatory gene of the CWI pathway, *MKK2* (AFUA_1G05800), also showed a significant change (Fig. 4C). These results suggest that the altered expression profiles may be associated with the phenotypic defects of the Δ*fabG* mutant. To validate the RNA-seq results, we analyzed selected genes by quantitative real-time PCR (qRT-PCR). The qRT-PCR results showed that the expression levels of these genes were largely consistent with the RNA-seq data (Fig. 4D). In summary, *fabG* may play a role in regulating the redox reaction and cell wall synthesis pathways by altering the Peroxisome and MAPK signaling pathways.

## Impact on oxidoreductases

In the Δ*fabG* mutant, SOD levels were significantly lower than in the WT (Fig. 5A), but CAT activity remained unchanged (Fig. 5B). Furthermore, since enzymes like SOD play a key role in scavenging intracellular ROS, further analysis of ROS levels revealed a significant increase in ROS content in the Δ*fabG* strain (Fig. 5C and D). Overall, these results indicate that the loss of *fabG* leads to a reduction in the activity of related oxidoreductases, including SOD, thereby impairing the ability to control oxidative stress.

## FabG is required for the maintenance of cell wall integrity in *A. fumigatus*

To assess the role of *fabG* in regulating CWI in *A. fumigatus*, CFW fluorescence staining was employed to quantify chitin content. The Δ*fabG* mutant exhibited a significant increase in chitin levels compared to the WT strain (Fig. 6A and B). Concurrently, β-glucan content was also markedly elevated (Fig. 6C). The susceptibility of Δ*fabG* to cell wall-perturbing agents was further evaluated. The mutant displayed reduced sensitivity to CR (which specifically binds β-glucan) and CFW (which targets chitin), whereas the complemented strain (Δ*fabG::fabG*⁺) displayed phenotypic restoration to WT levels under these stressors (Fig. 6D and E). Transmission electron microscopy revealed a pronounced thickening of the cell wall in Δ*fabG* (Fig. 6F and G). These findings collectively demonstrate that *fabG* plays a critical role in maintaining cell wall architecture and orchestrating the CWI pathway.

**TABLE 1** The impact of *fabG* on antifungal drug sensitivity[a]

| Strain | Antifungal susceptibility | | | | | | | | |
|---|---|---|---|---|---|---|---|---|---|
| | MICs or MEC (μg/mL) | | | | | | | | |
| | POS | ITR | VOR | CAS | ISA | CFW | CR | MD | $H_2O_2$ |
| WT | 0.125 | 0.5 | 0.125 | 0.125 | 0.25 | 25 | 25 | 6.25 | 0.03125% |
| Δ*fabG* | 0.125 | 0.5 | 0.25 | 0.5 | 0.25 | 200 | 400 | 50 | 0.25% |
| Δ*fabG::fabG*⁺ | 0.125 | 0.5 | 0.125 | 0.125 | 0.25 | 25 | 25 | 12.5 | 0.0625% |

[a]ISA, isavuconazole; MD, menadione; and $H_2O_2$, hydrogen peroxide.

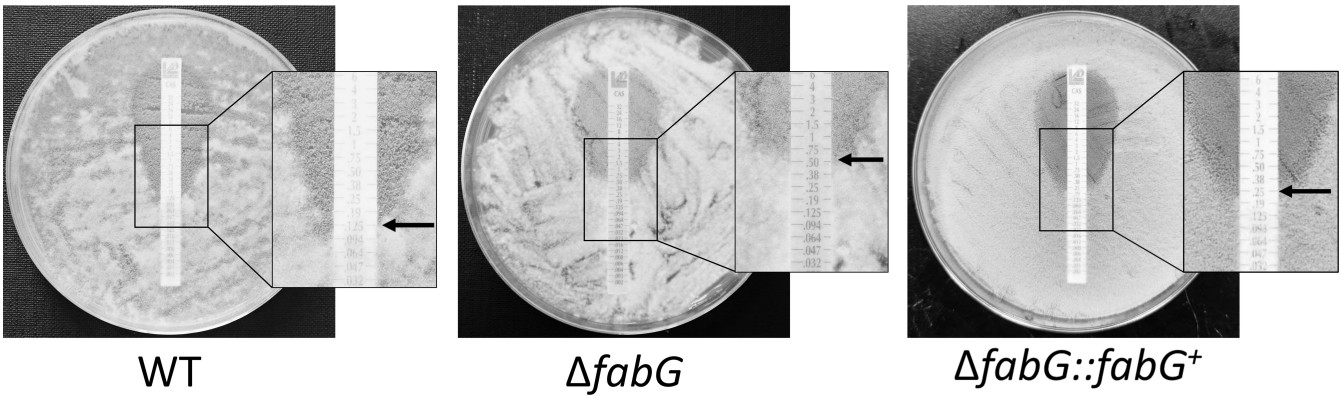

**FIG 3** Antifungal susceptibility using E-TEST gradient strips for caspofungin.

## Enhanced tolerance to osmotic pressure and oxidative stress in the *fabG* mutant

The Δ*fabG* mutant exhibits significantly enhanced tolerance to both osmotic and oxidative stress. Under oxidative stress conditions (H$_2$O$_2$, menadione) and high osmotic stress (NaCl, sorbitol, and mannitol), the growth inhibition of Δ*fabG* was only 14.47%, compared to 45.73% in WT and 46.81% in the Δ*fabG::fabG*$^+$ (Fig. 7). Although the baseline

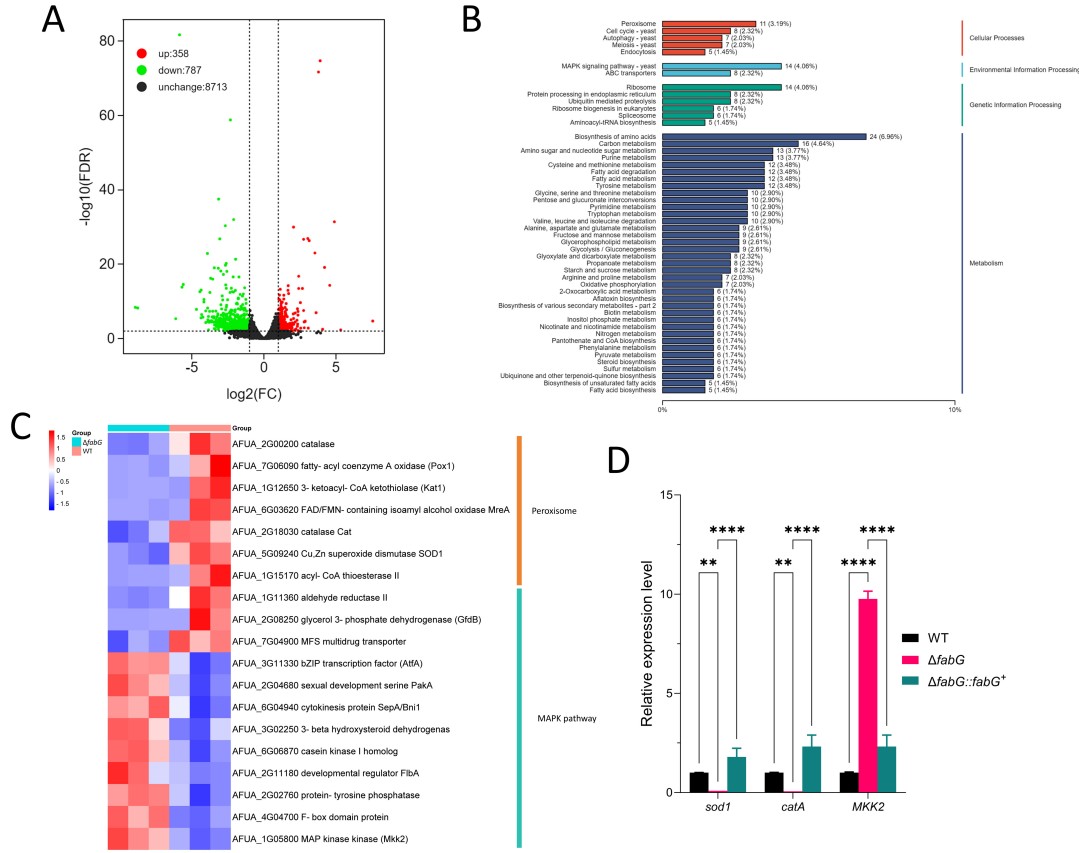

**FIG 4** Loss of *fabG* leads to changes in peroxisome and MAPK pathways. (A) The volcano plot showing the differentially expressed genes between the WT and Δ*fabG*. (B) KEGG pathway enrichment analysis of differentially expressed genes between the WT and Δ*fabG*. (C) A heatmap of the expression of peroxisome and MAPK pathway-related genes between the WT and Δ*fabG*. (D) The RT-qPCR analysis of selected genes in the WT, Δ*fabG*, and Δ*fabG::fabG*$^+$. The mRNA levels were normalized to those of the reference gene *tubA*. Statistical analysis was performed using two-way ANOVA with Tukey's test for multiple comparisons. **$P < 0.01$; ****$P < 0.0001$.

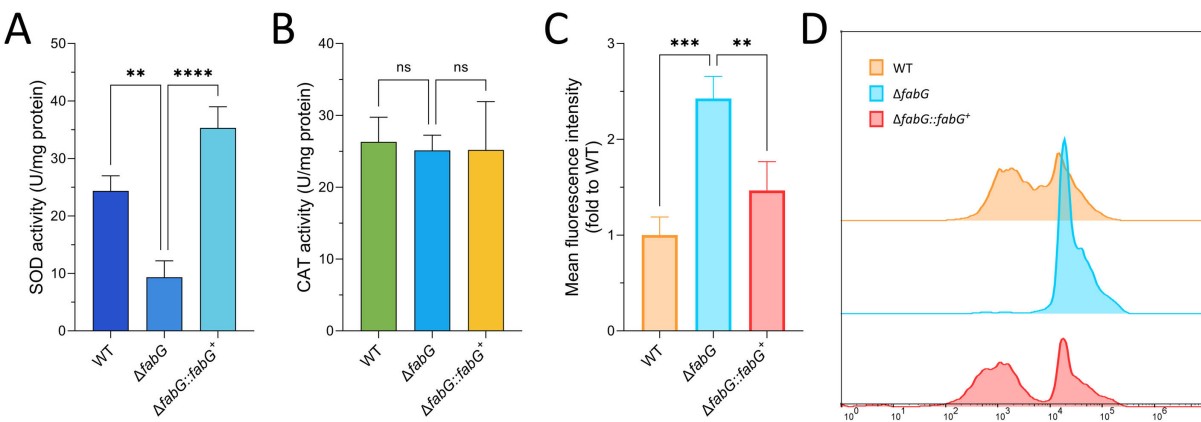

**FIG 5** Loss of *fabG* leads to decreased SOD activities and increased ROS levels. (A–C) Detection results for SOD activity, CAT activity, and ROS content. (D) Visualization of ROS levels using FlowJo version 10 software. Statistical analysis was performed using two-way ANOVA with Tukey's test for multiple comparisons. ns, $P > 0.05$; **$P < 0.01$; ***$P < 0.001$; and ****$P < 0.0001$.

growth rate of Δ*fabG* is somewhat reduced, its stability under stress conditions is significantly better than that of the control strains. The CWI pathway, which regulates stress tolerance, plays a critical role here, suggesting that the FabG protein may influence the fungal environmental adaptability through the regulation of the CWI pathway.

## The expression of CWI pathway genes under caspofungin/rapamycin treatment

As previously mentioned, the *fabG* gene plays a crucial role in regulating the integrity and thickness of the cell wall in *A. fumigatus*. The CWI pathway is a key regulatory pathway for fungal cell wall remodeling and synthesis, and *fabG* may influence the cell wall by participating in this process.

In this study, RNA sequencing analysis revealed that many differentially expressed genes were enriched in the MAPK pathway, with particular changes observed in the expression of the *MKK2* gene. This result was further validated by RT-qPCR. To explore this phenomenon in more detail, we treated the strains with CAS and rapamycin (Rapa) and analyzed the expression of key genes involved in the CWI pathway, including chitin synthase genes (*chsA*, *chsB*, *chsC*, *chsG*, and *csmB*), α-1,3-glucan synthase genes (*agsA* and *agsC*), β-1,3-glucan synthase genes (*fksA*), 1-3-β-glucan glucosyltransferase genes (*gelA* and *gelB*), and the CWI transcription factor (*rlmA*).

The results showed that, after CAS treatment, the transcription levels of CWI pathway target genes (*chsA*, *chsC*, *chsG*, *fksA*, *gelA*, *rlmA*, and *rlmB*) were significantly decreased in the WT strain, except for *gelB*, which showed reduced expression. In contrast, in the Δ*fabG* strain, only *chsG* expression decreased. Subsequently, after Rapa treatment, as expected, the expression levels of *chsA*, *chsC*, *fksA*, *gelA*, *rlmA*, *csmB*, *agsA*, and *agsC* were all increased, while *chsB* expression decreased. In the Δ*fabG* strain, however, only *chsG* expression increased.

Overall, whether co-cultured with CAS or Rapa, the downstream genes of the CWI pathway in the WT strain exhibited relatively strong changes in expression. For example, *rlmA* expression increased by 11-fold after Rapa treatment, while *fksA* expression decreased by 33-fold after CAS treatment. In contrast, the expression levels of the tested genes in the Δ*fabG* strain remained relatively stable, with changes generally within a twofold range. These results indicate that, in the presence of cell wall stressors, the CWI pathway signaling in Δ*fabG* remains stable, while the signaling in the WT strain undergoes significant changes.

Taken together, these findings suggest that the *fabG* gene may be a key effector gene in the CWI pathway. Its deletion likely causes a disruption in certain aspects of the CWI signaling, which in turn affects the regulation of cell wall synthesis and repair. This

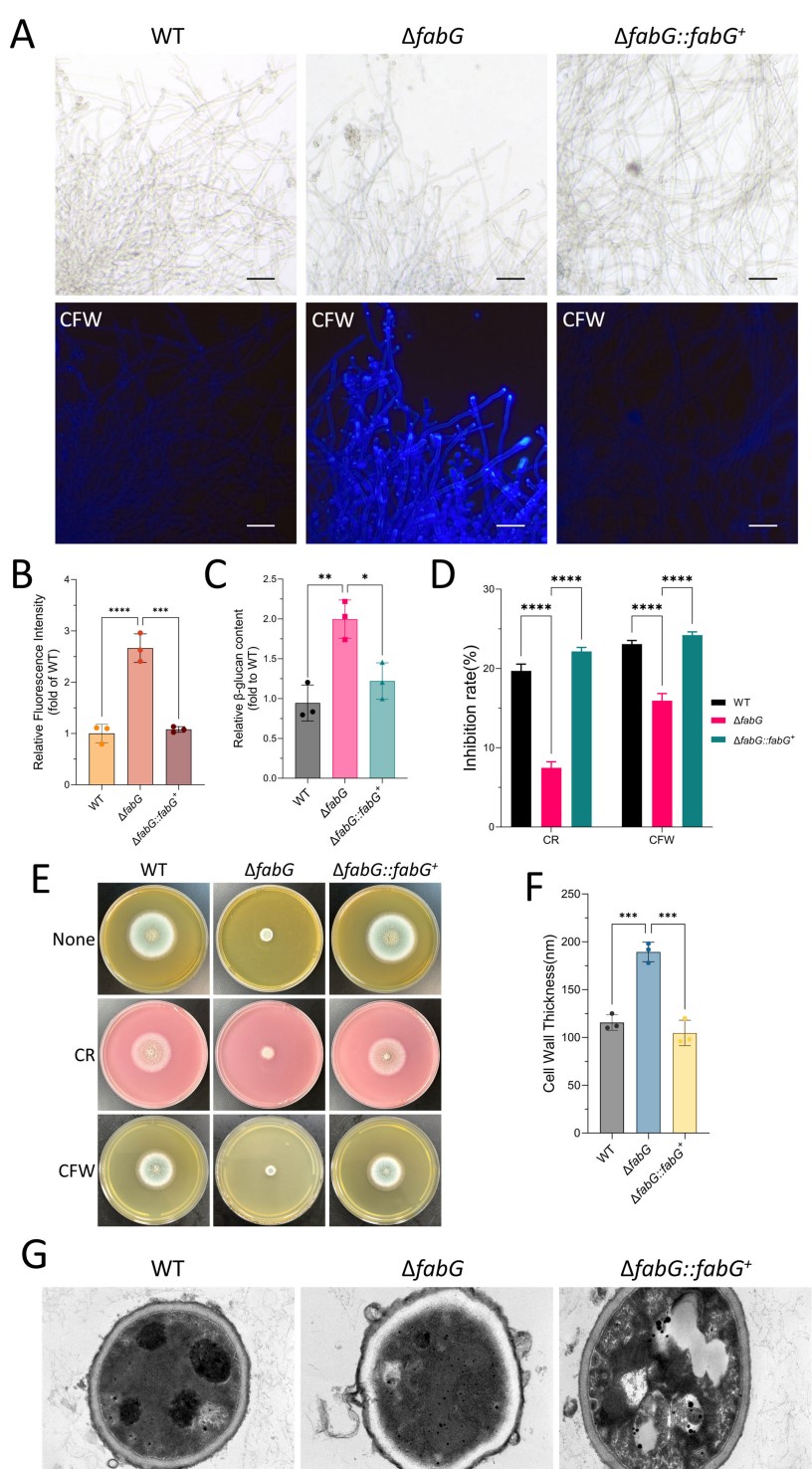

**FIG 6** *fabG* is required for the maintenance of cell wall integrity in *A. fumigatus*. (A) The distribution of the chitin content in the WT and Δ*fabG* mutants was visualized through fluorescence microscopy with CFW staining. Scale bar = 30 μm. (B) Fluorescence intensity analysis of CFW staining using ImageJ software, normalized to the WT. (C) β-glucan content. (D and E) Colony diameter and inhibition rate analysis after 48 hours of incubation on 100 μg/mL calcofluor white and 200 μg/mL Congo red media. The inhibition rate was calculated as follows: inhibition rate = (colony diameter without stressors − colony diameter with stressors)/colony diameter without stressors × 100%. The "colony diameter without stressors" refers to the colony diameter formed by strains cultured on non-supplemented solid SAB medium at 37°C for 48

Fig 6 (Continued)

hours and is labeled as "None" in the figure. (F) Quantification of cell wall thickness of the WT and Δ*fabG* mutants. (G) Representative transmission electron microscopy images of the WT and Δ*fabG* mutants. Scale bar = 200 nm. Statistical analysis in panels (B–D and F) was performed by two-way ANOVA with Tukey's test for multiple comparisons. ns, $P > 0.05$; *$P < 0.05$; **$P < 0.01$; ***$P < 0.001$; and ****$P < 0.0001$.

provides new insights into the role of *fabG* in the CWI pathway and highlights its importance in the maintenance of cell wall homeostasis.

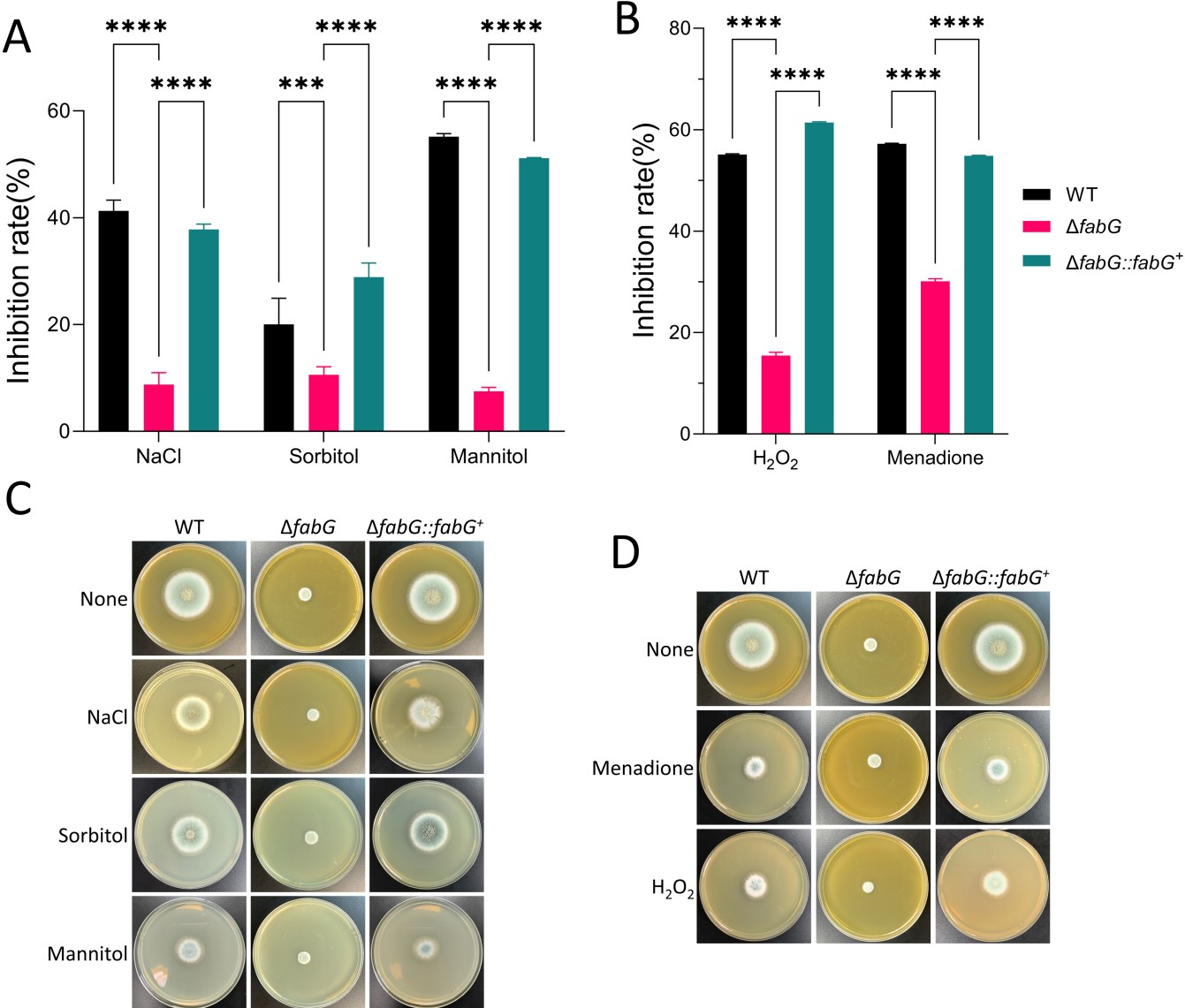

**FIG 7** Loss of *fabG* enhances adaptation to oxidative stress and osmotic pressure. (A and B) Relative hyphal growth inhibition of the indicated strains grown on solid SAB medium at 37°C for 48 hours. Statistical analyses were performed by two-way ANOVA with Tukey's test for multiple comparisons. (C) Colony morphology of the WT, Δ*fabG*, and complementation strains grown on solid SAB medium in the presence of 1 M NaCl, 1 M sorbitol, and 2 M mannitol at 37°C for 48 hours. (D) Colony morphology grown on solid SAB medium in the presence of 200 μM menadione and 5 M $H_2O_2$ at 37°C for 48 hours. The inhibition rate was calculated as follows: inhibition rate = (colony diameter without stressors − colony diameter with stressors)/colony diameter without stressors × 100%. The "colony diameter without stressors" refers to the colony diameter formed by strains cultured on non-supplemented solid SAB medium at 37°C for 48 hours and is labeled as "None" in the figure. ***$P < 0.001$; ****$P < 0.0001$.

## FabG enhances the macrophage killing activity against *A. fumigatus* conidia

Macrophages play a crucial role in antifungal immunity, such as phagocytosing and killing fungal spores (39). Therefore, we investigated the impact of FabG using an *in vitro* macrophage model. The results showed that Δ*fabG* conidia were more susceptible to phagocytosis and killing by macrophages. After 2 hours of incubation, a higher number of Δ*fabG* conidia were internalized (Fig. 8A and B). Following 4 hours of phagocytic killing, the remaining Δ*fabG* conidia were significantly fewer than those of the WT and complemented strains (Fig. 8C). Notably, after an extended 16-hour incubation, nearly all Δ*fabG* conidia were eradicated, whereas the WT and complemented strains exhibited visible hyphal growth and even conidial head formation (Fig. 8F).

TNF-α, IL-1β, and IL-6 are key inflammatory mediators secreted by macrophages upon exposure to *A. fumigatus* (40, 41). Thus, we also measured the expression levels of these cytokines. As expected, TNF-α, IL-1β, and IL-6 expression was significantly elevated in Δ*fabG*, particularly TNF-α, which showed a 38-fold increase compared to the WT (Fig. 8D). These findings suggest that Δ*fabG* is more easily recognized and eliminated by macrophages. To investigate whether *fabG* influences the virulence of *A. fumigatus*, we used the *G. mellonella* wax moth infection model to assess the virulence potential of the WT strain, Δ*fabG*, and Δ*fabG::fabG*$^+$. The Δ*fabG* exhibited a significantly reduced mortality rate of larvae compared to the WT, whereas the complementary strain showed similar mortality rates as the WT (Fig. 8E).

## DISCUSSION

Virulence is closely related to growth, which is particularly evident in *A. fumigatus*, where its growth rate often determines the degree of invasiveness (42–45). *A. fumigatus* can evade host immune surveillance through various mechanisms, such as modifying cell wall components to suppress host immune cell functions, thereby avoiding clearance by the immune system (14, 46–48). Oxidoreductases play a key role in eliminating intracellular ROS, thus protecting cells from oxidative damage (14, 49). These enzymes are considered the frontline defense in the cellular defense system (14). This study found that deletion of the *fabG* gene in *A. fumigatus* led to defects in colony growth and reduced SOD enzyme activity. In addition, the content of chitin and β-glucan in the cell wall increased, resulting in thicker cell walls, which contributed to enhanced stability of the strain under external stress. Moreover, the *G. mellonella* virulence model indicated that deletion of *fabG* may reduce the virulence and infectivity of *A. fumigatus*.

The fungal oxidative stress response involves enzymatic reactions from the thioredoxin and glutathione systems, including SOD and CAT (15, 50). SOD eliminates ROS by catalyzing the conversion of superoxide anions ($O_2^-$) into $H_2O_2$ and $O_2$, while CAT further degrades $H_2O_2$ into $H_2O$. *fabG*, *sodA*, and *catA* all belong to the oxidoreductase family and may be involved in oxidative stress defense in *A. fumigatus*, with *fabG* potentially regulating antioxidant enzyme activity indirectly by maintaining the NAD(P)H/NAD(P)$^+$ balance (14). Specifically, *sodA* and *catA* target ROS molecules without directly participating in cell wall metabolism, and deletion of their encoding genes is associated with sensitivity to oxidative stress. The putative SOD-encoding gene *sodD* is essential for survival; deletion of *sodA* or *sodB* leads to hypersensitivity to oxidative stress, while the *sodC* mutant only exhibits mild growth defects under heat stress (51). In contrast, *fabG* may function as a multifunctional oxidoreductase that not only regulates ROS homeostasis but also coordinates cell wall remodeling (such as β-glucan and chitin synthesis, Fig. 6) through modulation of the CWI pathway (Fig. 9). This results in enhanced tolerance to osmotic and cell wall stress—phenomena not previously reported in SOD or CAT mutants.

The *fabG* gene in *A. fumigatus* regulates its virulence and environmental adaptability through multiple mechanisms. A key function of oxidoreductases is the elimination of intracellular ROS (14), and *fabG*, as an oxidoreductase-related gene, similarly contributes to this process. Its deletion leads to reduced activity of oxidative enzymes such as SOD,

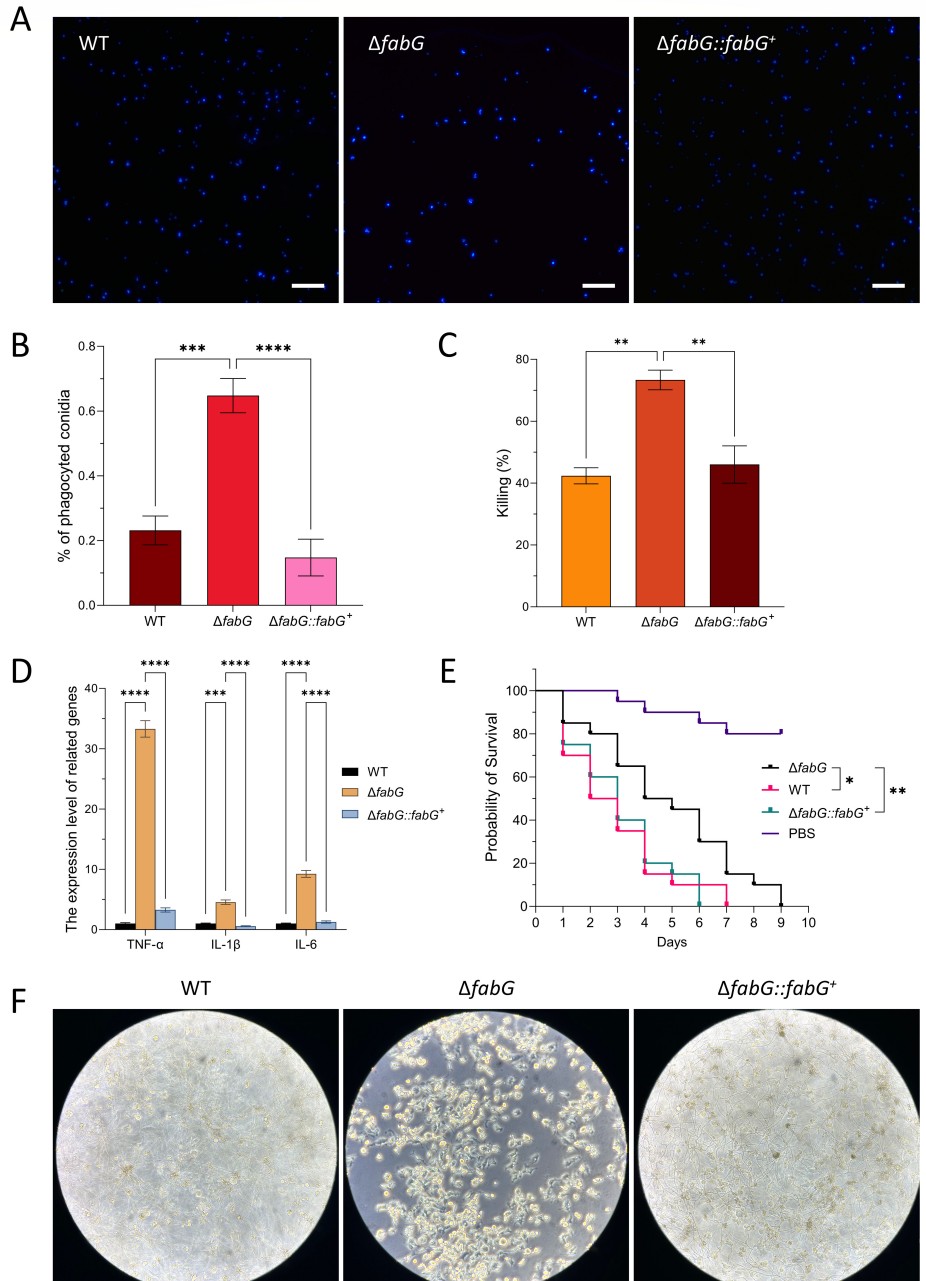

**FIG 8** FabG enhances the macrophage killing activity against *A. fumigatus* conidia. (A) CFW staining of non-phagocytosed conidia after 2 hours of co-culture with macrophages. Scale bar = 20 μm. (B) Phagocytosis index is increased in the Δ*fabG* (average ± standard deviation, $P ≤ 0.01$ compared to the WT and complemented strains). (C) Fungal killing assay: conidia were co-incubated with macrophages for 2 hours, after which non-phagocytosed conidia were removed. The cells were then further incubated for 4 hours before being plated onto SAB solid medium to assess conidial survival and calculate the killing rate. (D) Expression of inflammatory factors. Statistical analyses were performed by two-way ANOVA with Tukey's test for multiple comparisons. (E) Survival curves for *G. mellonella* larvae infected with the WT, Δ*fabG*, and Δ*fabG::fabG*+. Each experiment was replicated three times independently. Statistical analysis was performed using the log-rank test. (F) Microscopic observation of conidia after 16 hours of co-culture with macrophages. Scale bar = 20 μm. **$P < 0.01$; ***$P < 0.001$; and ****$P < 0.0001$.

resulting in ROS accumulation (52), suggesting that *fabG* may also play a role in counteracting ROS generation. As highly reactive molecules, ROS can induce oxidative stress (52), and their accumulation—along with redox imbalance—can further activate the

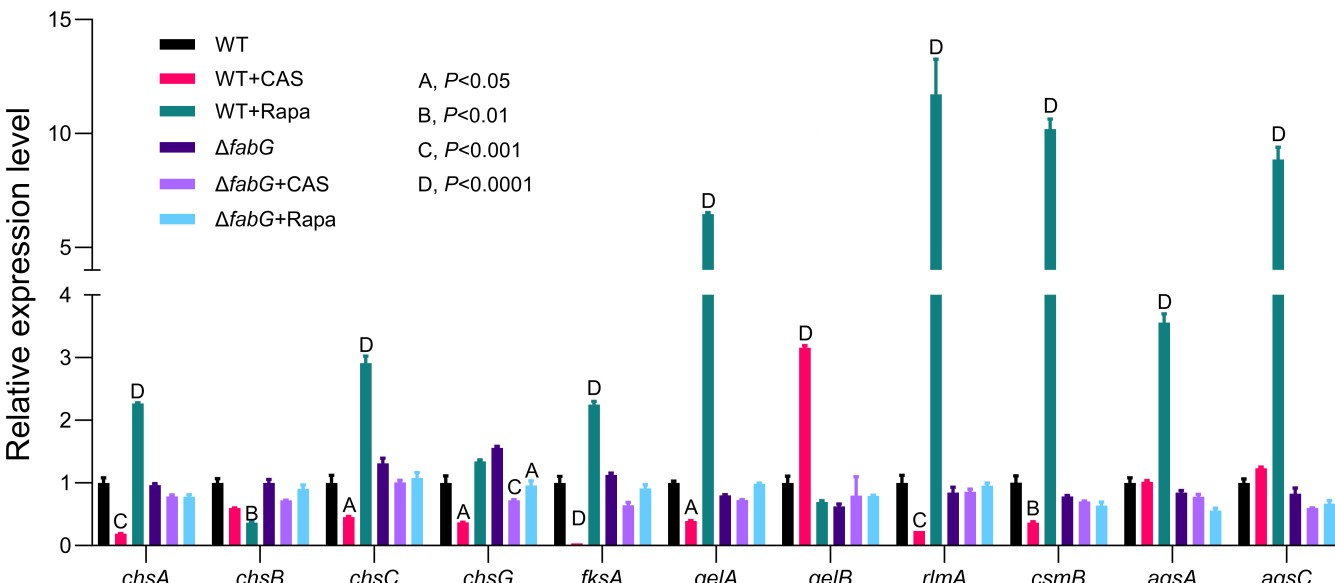

**FIG 9** Expression levels of genes related to the CWI pathway in *A. fumigatus*. Relative expression levels of cell wall synthesis genes following caspofungin (2 µg/mL) and rapamycin (0.5 µg/mL) treatment for 5 hours. Gene expression was measured by RT-qPCR and normalized to the *tubA* gene, with normalization to WT. Expression was calculated using the $2^{-\Delta\Delta CT}$ method. Statistical analysis was performed using two-way ANOVA, and Tukey's test was applied for multiple comparisons correction. A, $P < 0.05$; B, $P < 0.01$; C, $P < 0.001$; and D, $P < 0.0001$.

cascade of the CWI signaling pathway (17). The biosynthesis of the *A. fumigatus* cell wall is regulated by the CWI pathway, which amplifies external signals through a signaling cascade and mediates downstream metabolic responses, especially during environmental stress, to regulate cell wall formation and remodeling (53). For example, the CWI pathway is the main signaling pathway controlling environmental stress response and cell wall component synthesis in *Saccharomyces cerevisiae* (17, 54). To counter persistent ROS, the fungal cell wall may undergo adaptive remodeling through compensatory thickening. In the absence of *fabG*, the thickened cell wall exhibited enhanced resistance to stressors such as oxidative and osmotic stress. This compensatory thickening, accompanied by abnormal increases in chitin and β-glucan, led to reduced sensitivity to cell wall-targeting antifungal drugs such as CAS. The CWI pathway is also critical for fungal virulence; for instance, the growth and virulence of *Fusarium graminearum* are highly dependent on it (55). Similarly, the thickened cell wall structure of the Δ*fabG* mutant results in slower growth, likely due to the need for reallocating energy and resources toward cell wall remodeling—a mechanism that may enhance fungal survival within the host, ultimately manifesting as reduced growth rate in the Δ*fabG* strain. Studies have shown that the fungal CW plays multiple roles in virulence, and *A. fumigatus* mutants with defective CWI show significantly attenuated virulence (54). Our study observed a similar phenomenon: the thickened cell wall exposed more immunogenic components such as β-1,3-glucan, making the fungus more recognizable and susceptible to immune cell attack. This may explain the reduced virulence caused by *fabG* deletion.

The cell wall of *A. fumigatus* plays a dual role: it not only maintains structural integrity but also serves as a critical immunological interface that mediates host–pathogen interactions (32, 56–58). Its composition directly influences fungal virulence and shapes the host immune response. During infection, pattern recognition receptors on host immune cells—such as Dectin-1—recognize β-1,3-glucan in the fungal cell wall, triggering downstream immune signaling cascades (59–62). Notably, increased surface exposure of β-1,3-glucan significantly enhances the phagocytosis of conidia by macrophages (63). Swollen conidia or early germ tubes, which expose large amounts of β-1,3-glucan, can strongly activate the Dectin-1 pathway and induce robust secretion

of proinflammatory cytokines such as IL-1β and TNF-α (64). *fabG* may influence the structure and composition of the *A. fumigatus* CW by regulating the expression of genes involved in chitin and β-glucan biosynthesis. The elevated β-1,3-glucan content observed in the *fabG* mutant is associated with enhanced macrophage-mediated phagocytosis and fungal killing, suggesting that *fabG* may contribute to immune recognition and modulation during IA. This study found that the thickening of the *A. fumigatus* cell wall following *fabG* gene deletion is closely associated with the activation of the CWI pathway. Moreover, the increased resistance of the mutant to CAS may interfere with the recognition of pathogen-associated molecular patterns by host pattern recognition receptors. In summary, this study confirms that *fabG* likely plays a key role in the growth and virulence of *A. fumigatus* by regulating oxidoreductase activity. Additionally, *fabG* affects fungal sensitivity to cell wall-disrupting agents and cell wall thickness by modulating the expression of genes related to the CWI pathway, thereby coordinating fungal growth and virulence. However, current evidence is limited to molecular and basic phenotypic observations. The specific interactions between *fabG* and oxidoreductases such as SOD and CAT require further validation using methods such as yeast two-hybrid assays or co-immunoprecipitation. The connection between *fabG* and the CWI pathway also lacks definitive evidence—for example, confirmation via western blot using phospho-specific antibodies or phospho-proteomics would be necessary. It is also possible that the newly observed functions in the gene knockout strain arise from interactions between *fabG* deletion and KU80 deficiency. Overall, *fabG* exhibits novel functions and may play a role in the oxidoreductase system, representing a potential therapeutic target for combating *A. fumigatus* infections in the future.

## ACKNOWLEDGMENTS

The authors thank everyone who contributed to the success of this research, including colleagues, institutions, and funding bodies.

This work was supported by the Jingzhou Science and Technology Plan Project (grant number 2024HD34); the Yangtze University Science and Technology Aid to Tibet Medical Talent Training Program Project (grant number 2023YZ06); and the Key Research and Development program of Hubei Province (grant number 2024BCB043).

All authors contributed to the research in this report. Y.S. and H.Z. conceptualized the study. H.N. designed the methodology. X.T. and T.C. helped with software. Y.Z. validated the study. X.Z. performed formal analysis. Y.C. performed the investigation. Y.S. provided the resources. H.N. curated the data. H.Z. wrote the original draft. Y.S. and H.Z. reviewed and edited the manuscript. Y.Z. and M.P. visualized the study. X.Z. supervised the study. Y.S. contributed to project administration. Y.S. and H.Z. acquired funding. All authors have read and agreed to the published version of the manuscript.

## AUTHOR AFFILIATIONS

[1]Department of Dermatology, Hubei Provincial Clinical Research Center for Diagnosis and Therapeutics of Pathogenic Fungal Infection, Jingzhou Hospital Affiliated to Yangtze University, Jingzhou, Hubei Province, China

[2]Department of Nephrology, Hubei Provincial Clinical Research Center for Diagnosis and Therapeutics of Pathogenic Fungal Infection, Jingzhou Hospital Affiliated to Yangtze University, Jingzhou, Hubei Province, China

[3]Department of Clinical Medicine, Yangtze University, Jingzhou, Hubei Province, China

[4]Department of Otolaryngology, Hubei Provincial Clinical Research Center for Diagnosis and Therapeutics of Pathogenic Fungal Infection, Jingzhou Hospital Affiliated to Yangtze University, Jingzhou, Hubei Province, China

## AUTHOR ORCIDs

Heng Zhang http://orcid.org/0009-0002-6618-5311
Yi Sun http://orcid.org/0000-0002-4489-3803

## FUNDING

| Funder | Grant(s) | Author(s) |
|---|---|---|
| Jingzhou Science and Technology Plan Project | 2024HD34 | Heng Zhang |
| Yangtze University Science and Technology Aid to Tibet Medical Talent Training Program Project | 2023YZ06 | Yi Sun |
| Key Research and Development Plan of Hubei Province | 2024BCB043 | Yi Sun |

## AUTHOR CONTRIBUTIONS

Heng Zhang, Conceptualization, Formal analysis, Funding acquisition, Methodology, Project administration, Resources, Software, Writing – original draft | Hua Ni, Investigation, Project administration, Software, Validation | Xinyi Tao, Project administration, Validation, Writing – original draft | Tian Chen, Methodology, Resources, Supervision, Visualization, Writing – original draft | Yi Zhang, Methodology, Project administration, Resources | Xiaolei Zhu, Methodology, Resources, Supervision | Yinping Chen, Project administration, Supervision | Mengqi Peng, Methodology, Project administration, Validation | Yi Sun, Conceptualization, Data curation, Formal analysis, Funding acquisition, Software, Supervision, Writing – original draft, Writing – review and editing

## DATA AVAILABILITY

All RNA-seq data files are available at the NCBI Sequence Read Archive (SRA) database under the accession number PRJNA1233255 for WT, Δ*fabG* of *A. fumigatus* conidia RNA-seq.

## ADDITIONAL FILES

The following material is available online.

### Supplemental Material

**Supplemental Material (Spectrum02092-25-s0001.docx).** Tables S1 and S2; Fig. S1 and S2.

### Open Peer Review

**PEER REVIEW HISTORY (review-history.pdf).** An accounting of the reviewer comments and feedback.

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
