## [Reviewer comments · Microbiology Spectrum]

Microbiology Spectrum

Oxidoreductase gene *fabG* contributes to fungal development, cell wall integrity, and virulence in *Aspergillus fumigatus*

Heng Zhang, Hua Ni, Xinyi Tao, Tian Chen, Yi Zhang, Xiaolei Zhu, Yinping Chen, Mengqi Peng, and Yi Sun

Corresponding Author(s): Yi Sun, Jingzhou Hospital Affiliated to Yangtze University

Review Timeline:

Submission Date:	July 11, 2025
Editorial Decision:	August 8, 2025
Revision Received:	September 14, 2025
Accepted:	October 4, 2025

Editor: Vanessa Varaljay

Reviewer(s): Disclosure of reviewer identity is with reference to reviewer comments included in decision letter(s). The following individuals involved in review of your submission have agreed to reveal their identity: Hee-Soo Park (Reviewer #2)

Transaction Report:

DOI: <https://doi.org/10.1128/spectrum.02092-25>

Re: Spectrum02092-25 (**Oxidoreductase gene *fabG* contributes to fungal development, cell wall integrity, and virulence in *Aspergillus fumigatus***)

Dear Dr. Yi Sun:

Thank you for the privilege of reviewing your work. Below you will find my comments, instructions from the Spectrum editorial office, and the reviewer comments.

Please carefully address the reviewer's comments particularly with respect to the presentation of the data and the figures.

Revision Guidelines

Sincerely,
Vanessa Varaljay
Editor
Microbiology Spectrum

Reviewer #1 (Comments for the Author):

This study conducts a rigorous and impactful investigation into the function of the *fabG* gene in *Aspergillus fumigatus*, filling a critical gap in our understanding of this clinically significant pathogen. Employing a comprehensive, multi-layered approach that includes gene editing, phenotypic analysis, biochemical assays, and molecular-level research, it yields rich and consistent data, strongly confirming that *fabG* is a key gene regulating fungal development, redox homeostasis, cell wall integrity (CWI), and

virulence. Overall, this is an interesting study, and I recommend it for publication. My comments:

1. Supplement the sources of RAW 264.7 macrophages and KU80 strain.
2. Label the unit on the Y-axis in Fig. 2G.
3. Use "panels" in line 437.
4. Abbreviate "cell wall integrity" as "CWI" in line 179.
5. Specify the formula for growth inhibition rate in Fig. 6 and Fig. 7.

Reviewer #2 (Public repository details (Required)):

They conducted RNA-seq analysis.

"All RNA seq data files are available at the NCBI Sequence Read Archive (SRA) database under the accession number PRJNA1233255 for WT, Δ fabG of *A. fumigatus* conidia RNA seq"

Reviewer #2 (Comments for the Author):

This manuscript investigates the function of the fabG gene, which encodes an oxidoreductase, in the pathogenic fungus *Aspergillus fumigatus*. The authors generated a deletion and complemented strains and performed phenotypic and gene expression analyses to elucidate the role of FabG. Their results demonstrate that FabG is involved not only in asexual development but also in the cell wall integrity (CWI) pathway and oxidative stress response. Additionally, macrophage assays revealed that the fabG deletion mutant exhibits reduced virulence. These findings contribute valuable insights into the pathogenic mechanisms of *A. fumigatus*.

I would like to offer the following suggestions to further improve the manuscript:

- The virulence data in Figure 2I may be better presented alongside the macrophage assay results to provide a more comprehensive interpretation.
- Figure 1A could be enlarged for better readability, and the gene information used in this figure should be clearly described in the Materials and Methods or Figure Legend.
- Figure 2J should be revised for clarity, as the data presented are currently difficult to interpret.
- Please include the results of the complemented strains in Figure 3 for proper comparison.
- Consider switching the order of Figures 4A and 4B to improve logical flow.
- The results of the complemented strains should also be presented in Figure 4D.
- In Figures 6E and 7C/D, please include control media results (e.g., CZA or SAB), keeping in mind the possibility of overlapping data across figures.
- In Figure 6G, I recommend presenting the data in the order of WT, fabG, and fabG::fabG⁺ for better comparison.
- Lines 311 & 312, replace "fabG" with "FabG"
- Line 321, italic " Δ fabG"
- Line 360 replace "fabG" with "FabG"

Dear Editor,

We are very grateful for your constructive comments and suggestions for our manuscript entitled "Oxidoreductase gene *fabG* contributes to fungal development, cell wall integrity, and virulence in *Aspergillus fumigatus*"(Spectrum02092-25). Your comments are very valuable and helpful for improving our manuscript. In the following, the responses to all the comments are provided one by one.

Reviewer #1 (Comments for the Author):

This study conducts a rigorous and impactful investigation into the function of the *fabG* gene in *Aspergillus fumigatus*, filling a critical gap in our understanding of this clinically significant pathogen. Employing a comprehensive, multi-layered approach that includes gene editing, phenotypic analysis, biochemical assays, and molecular-level research, it yields rich and consistent data, strongly confirming that *fabG* is a key gene regulating fungal development, redox homeostasis, cell wall integrity (CWI), and virulence. Overall, this is an interesting study, and I recommend it for publication. My comments:

1. Supplement the sources of RAW 264.7 macrophages and KU80 strain.

Answer: Thank you for your suggestions. I have supplemented the sources of RAW 264.7 macrophages and the KU80 strain.

2. Label the unit on the Y-axis in Fig. 2G.

Answer: Thank you for your suggestion. I have supplemented the unit of the y-axis in Fig. 2G as "spores/mL".

3. Use "panels" in line 437.

Answer: Thank you for your suggestion. I have revised "panel" to "panels" here.

4. Abbreviate "cell wall integrity" as "CWI" in line 179.

Answer: Thank you for your suggestion. I have abbreviated "cell wall integrity" to CWI here.

5. Specify the formula for growth inhibition rate in Fig. 6 and Fig. 7.

Answer: Thank you for your suggestion. I have supplemented the calculation formula for the inhibition rate here.

Reviewer #2 (Public repository details (Required)):

They conducted RNA-seq analysis.

"All RNA seq data files are available at the NCBI Sequence Read Archive (SRA) database under the accession number PRJNA1233255 for WT, Δ *fabG* of *A. fumigatus* conidia RNA seq"

Answer: Thank you for your careful review and confirmation of our RNA-seq data deposition. We confirm that all RNA-seq data of *A. fumigatus* WT and Δ *fabG* conidia have been deposited in the NCBI Sequence Read Archive (SRA) under the accession number PRJNA1233255, and the data are publicly accessible. Additionally, we have provided a statement in the Declarations section: "Availability of data and materials: All RNA-seq data files are available at the NCBI Sequence Read Archive (SRA) database under the accession number PRJNA1233255 for WT, Δ *fabG* of *A. fumigatus* conidia RNA-seq."

Reviewer #2 (Comments for the Author):

This manuscript investigates the function of the *fabG* gene, which encodes an oxidoreductase, in the pathogenic fungus *Aspergillus fumigatus*. The authors generated a deletion and complemented strains and performed phenotypic and gene expression analyses to elucidate the role of FabG. Their results demonstrate that FabG is involved not only in asexual development but also in the cell wall integrity (CWI) pathway and oxidative stress response. Additionally, macrophage assays revealed that the *fabG* deletion mutant exhibits reduced virulence. These findings contribute valuable insights into the pathogenic mechanisms of *A. fumigatus*.

I would like to offer the following suggestions to further improve the manuscript:

1. The virulence data in Figure 2I may be better presented alongside the macrophage assay results to provide a more comprehensive interpretation.

Answer: Thank you for your suggestion. I have presented the data from Figure 2I together with the results of the phagocytosis assay.

2. Figure 1A could be enlarged for better readability, and the gene information used in this figure should be clearly described in the Materials and Methods or Figure Legend.

Answer: Thank you for your suggestion. I have appropriately enlarged Fig. 1A and added detailed explanations of the gene information in the figure legend.

3. Figure 2J should be revised for clarity, as the data presented are currently difficult to interpret.

Answer: Thank you for your suggestion. I have enlarged Fig. 2J and used a clearer image.

4. Please include the results of the complemented strains in Figure 3 for proper comparison.

Answer: Thank you for your suggestion. I have included the results of the complementation strain in Figure 3.

5. Consider switching the order of Figures 4A and 4B to improve logical flow.

Answer: Thank you for your suggestion. I have swapped the order of 4A and 4B.

6. The results of the complemented strains should also be presented in Figure 4D.

Answer: Thank you for your suggestion. I have added the results of the complementation strain in 4D.

7. In Figures 6E and 7C/D, please include control media results (e.g., CZA or SAB), keeping in mind the possibility of overlapping data across figures.

Answer: Thank you for your suggestion. Both Figure 6E and Figures 7C/D were cultured on 90mm culture media, with the colony diameter after 48 hours of cultivation on SAB medium used as the control. I have supplemented the images of the control medium in the figures.

8. In Figure 6G, I recommend presenting the data in the order of WT, *fabG*, and *fabG::fabG⁺* for better comparison.

Answer: Thank you for your suggestion. I have presented the data in the order of WT, $\Delta fabG$, and $\Delta fabG::fabG^+$.

8. Lines 311 & 312, replace "fabG" with "FabG"

Answer: Thank you for your suggestion. I have replaced "fabG" with "FabG".

9. Line 321, italic " Δ fabG"

Answer: Thank you for your suggestion. I have italicized " Δ fabG".

10. Line 360 replace "fabG" with "FabG"

Answer: Thank you for your suggestion. I have replaced "*fabG*" with "FabG".

Sincerely,

Yi Sun

Department of Dermatology, Jingzhou Hospital Affiliated to Yangtze University,
Hubei Provincial Clinical Research Center for Diagnosis and Therapeutics of
Pathogenic Fungal Infection, Jingzhou, Hubei Province, 434100, China

Tel:+86 18071883358

E-mail:jzzyysy@163.com

Re: Spectrum02092-25R1 (**Oxidoreductase gene *fabG* contributes to fungal development, cell wall integrity, and virulence in *Aspergillus fumigatus***)

Dear Dr. Yi Sun:

Thank you for your careful attention to the reviewers' comments and for improving the manuscript accordingly.

Your manuscript has been accepted, and I am forwarding it to the ASM production staff for publication. Your paper will first be checked to make sure all elements meet the technical requirements. ASM staff will contact you if anything needs to be revised before copyediting and production can begin. Otherwise, you will be notified when your proofs are ready to be viewed.

Sincerely,
Vanessa Varaljay
Editor
Microbiology Spectrum

Reviewer #2 (Comments for the Author):

The authors addressed all issues raised by reviewers